# Germacrene A Synthases for Sesquiterpene Lactone Biosynthesis Are Expressed in Vascular Parenchyma Cells Neighboring Laticifers in Lettuce

**DOI:** 10.3390/plants11091192

**Published:** 2022-04-28

**Authors:** Moonhyuk Kwon, Connor L. Hodgins, Tegan M. Haslam, Susan A. Roth, Trinh-Don Nguyen, Edward C. Yeung, Dae-Kyun Ro

**Affiliations:** 1Department of Biological Sciences, University of Calgary, Calgary, AL T2N 1N4, Canada; mkwon@gnu.ac.kr (M.K.); clhodgin@ucalgary.ca (C.L.H.); tegan.haslam@biologie.uni-goettingen.de (T.M.H.); susan.roth1@ucalgary.ca (S.A.R.); don.nguyen@ubc.ca (T.-D.N.); yeung@ucalgary.ca (E.C.Y.); 2Division of Applied Life Sciences (BK21 Plus), Plant Molecular Biology and Biotechnology Research Center, Gyeongsang National University, Jinju 660-701, Korea

**Keywords:** lettuce, laticifers, isoprenoid, sesquiterpene lactone, natural rubber

## Abstract

Sesquiterpene lactone (STL) and natural rubber (NR) are characteristic isoprenoids in lettuce (*Lactuca sativa*). Both STL and NR co-accumulate in laticifers, pipe-like structures located along the vasculature. NR-biosynthetic genes are exclusively expressed in laticifers, but cell-type specific expression of STL-biosynthetic genes has not been studied. Here, we examined the expression pattern of germacrene A synthase (*LsGAS*), which catalyzes the first step in STL biosynthesis in lettuce. Quantitative PCR and Illumina read mapping revealed that the transcripts of two *GAS* isoforms (*LsGAS1*/*LsGAS2*) are expressed two orders of magnitude (~100–200) higher in stems than laticifers. This result implies that the cellular site for *LsGAS1*/*2* expression is not in laticifers. To gain more insights, promoters of *LsGAS1*/*2* were cloned and fused to β-glucuronidase (*GUS*), followed by transformations of lettuce with these promoter-*GUS* constructs. In in situ GUS assays, the *GUS* expression driven by the *LsGAS1/2* promoters was tightly associated with vascular bundles. High-resolution microsections showed that GUS signals are not present in laticifers but are detected in the vascular parenchyma cells neighboring the laticifers. These results suggest that expression of *LsGAS1/2* occurs in the parenchyma cells neighboring laticifers, while the resulting STL metabolites accumulate in laticifers. It can be inferred that active metabolite-trafficking occurs from the parenchyma cells to laticifers in lettuce.

## 1. Introduction

Lettuce (*Lactuca sativa*) is a major leafy vegetable crop from the Asteraceae family with an annual production of ~27 million tonnes [1]. Aside from its agricultural value, lettuce has attracted attention for two specialized metabolites—natural rubber (NR; *cis*-1,4-polyisoprene) and sesquiterpene lactones (STLs), a group of fifteen carbon isoprenoids with a characteristic α-methylene γ-lactone moiety [2,3]. NR is a commodity with which hundreds of industrial products are manufactured, including tyres [4]. STLs exhibit diverse bioactivities, such as parthenolide as an anti-inflammatory agent known to inhibit NF-κB activation [5]. Both NR and STLs are isoprenoid natural products derived from isopentenyl diphosphate (IPP) and co-accumulate in laticifer cells [2,3]. Laticifers are a group of cells linearly fused with one another through perforated cell walls to form a network of multicellular structures along the vasculature [6]. Latex, the cytoplasmic content of laticifers, is metabolically active to synthesize diverse specialized metabolites. In lettuce, NR and STLs are released presumably as defensive chemicals when laticifers are ruptured by physical damage during herbivory.

Among the hundreds of natural products in laticifers, the most notable example is NR from the Para rubber tree (*Hevea brasiliensis*) [7]. The metabolic competency of laticifers can be shown by the massive production of NR in Para rubber tree (~13 million tonnes per year by the International Rubber Study Group’s report). However, in-depth molecular studies of laticifers in plants, including Para rubber tree, are difficult due to recalcitrance to transformation, the perennial nature of woody plants, and/or obligate outcrossing. Several model plants (i.e., Arabidopsis, poplar, rice, alfalfa) are not suitable for these studies, as they do not develop laticifers. To overcome these problems, we have initiated molecular studies of lettuce laticifers since lettuce is a transformable, annual, diploid, and self-pollinating plant [8,9]. Furthermore, the recent release of lettuce genomic data further supports molecular genetic studies of lettuce laticifers [10].

Although NR biosynthesis remains to be fully understood, it was reported that a protein complex is formed by *cis*-prenyltransferase isoform 3 (LsCPT3) and CPT-Like protein isoform 2 (LsCPTL2) on the cytosolic side of the endoplasmic reticulum (ER) in lettuce (Figure 1A) [8], and its related species dandelion also has orthologous proteins that form a protein complex on the ER [11]. The original gene name *LsCPTL2* has been renamed *LsCBP2* (*Lactuca sativa CPT-binding protein 2*) [12], and hereafter, we will use *LsCBP2* to refer to *LsCPTL2*. RNAi-silencing showed that the protein complex formed by LsCPT3 and LsCBP2 is required for NR biosynthesis in lettuce [8], and qPCR data and *LsCPT3/LsCBP2* promoter analysis by reporter gene fusion showed that these two genes are highly and exclusively expressed in lettuce laticifers [8,13]. Based on phylogenetic analysis, *LsCPT3/LsCBP2* for NR biosynthesis seems to have diverged from the primary metabolic genes, *LsCPT1/LsCBP1* [8]. The protein pair, LsCPT1/LsCBP1, also forms a protein complex on the ER and synthesizes dehydrodolichyl diphosphate (DHDD), which is transformed into a group of oligomeric isoprene polymers, dolichol [8,14,15,16] (Figure 1A). Dolichol serves as a carrier molecule of sugars for protein glycosylation in eukaryotes [17] and for peptidoglycan cell walls in prokaryotes [18], making it an indispensable metabolite in both domains of life. Therefore, lettuce has evolved to produce two types of *cis*-1,4-polyisoprenes, dolichol and NR.

STLs also accumulate in lettuce latex, and their biosynthesis in lettuce is characterized down to costunolide (Figure 1A) [19,20,21]. The first reaction is catalyzed by germacrene A synthase (*LsGAS*), which synthesizes germacrene A from farnesyl diphosphate (FPP). Sequential oxygenations of germacrene A by two cytochrome P450s (germacrene A oxidase and costunolide synthase) yield a core STL precursor, costunolide. Despite our advanced understanding of STL biosynthesis, cell-type specific expression of STL-biosynthetic genes has not been studied in lettuce. Since both NR and STLs co-accumulate in high quantities in laticifers, it has been assumed that both of their biosynthetic genes are simultaneously expressed in lettuce laticifers.

Here, the occurrence of *LsGAS* transcripts was analyzed by quantitative PCR (qPCR) and targeted Illumina read mapping in different tissues. We further examined *LsGAS* promoter activities by β-glucuronidase (*GUS*)-fusion using a transgenic approach. Unexpectedly, no evidence for the laticifer-specific expression of *LsGAS* was obtained. In situ GUS staining patterns showed that the *LsGAS* promoter drives gene expression in the parenchyma cells neighboring the laticifers. NR and STLs co-accumulate in laticifers, but their respective genes for biosynthesis are expressed in distinct cells, highlighting the importance of cellular compartmentalization in isoprenoid (STL and NR) metabolism in lettuce and possibly other plants.

## 2. Results

### 2.1. Assessing Relative LsGAS Transcripts by qPCR Analysis

Although previously reported [2,3], we cultivated lettuce in a phytochamber and confirmed the presence of STLs and NR in lettuce latex by LC-quadruple time-of-flight (qTOF) and HPLC (high performance liquid chromatography)-GPC (gel permeation column)-ELSD (evaporative light scattering detector), respectively. Three major STLs with mass accuracy of less than 5 Δppm, and NR with an average Mw of 2.3 million Da, were detected in lettuce latex (Figure 1B,C). 

The first reaction for STL biosynthesis is catalyzed by germacrene A synthase (*GAS*). In lettuce and its related species chicory, three *GAS* isoforms have been biochemically characterized [22,23]. These *GAS* isoforms are referred to as *LsGAS1-3*. Genomic analysis of lettuce with Phytozome (phytozome-next.jgi.doe.gov, accessed on 14 April 2022) identified the three genes in three genomic loci (*LsGAS1* in *Lsat_1_v5_gn_8_116421.1*; *LsGAS2* in *Lsat_1_v5_gn_8_116340.1*, and *LsGAS3* in *Lsat_1_v5_gn_2_29221.1*). *LsGAS1* and *LsGAS2* are clustered into linkage group #8 with a distance of 10.9 kb (Figure 2A), while *LsGAS3* is located in linkage group #2. *LsGAO1* is also linked with *LsGAS1/2*, occurring 226 kb upstream of *LsGAS2* (Figure 2A). Two gene models were identified between the *LsGAO1* and *LsGAS1/2* loci; one has no annotation (*Lsat_1_v5_gn_8_116301.1*) and the other was annotated as an Arabidopsis broad-spectrum mildew resistance protein RPW8 (*Lsat_1_v5_gn_8_115321.1*). Both have no functional relation to STL biosynthesis. We further examined 220 kb upstream of *LsGAO1* and 650 kb downstream of *LsGAS1* in the lettuce genome. However, no gene model with a possible role in STL biosynthesis could be identified. Therefore, we concluded that *LsGAS1/2* and their immediate downstream metabolic gene, *LsGAO1*, form a gene cluster in linkage group #8 in lettuce, but no evidence for extensive gene clustering involving other downstream genes was obtained. 

To evaluate the expression of *LsGAS1-3* in lettuce, we first examined their relative transcripts in latex and other tissues with qPCR. Relatively pure liquid latex was isolated by making an incision in the lettuce stem. In the qPCR analysis, *LsGAS1*/2 displayed co-relating expression patterns in six different tissues, but the expression of *LsGAS3* differed from those of *LsGAS1/2*. Unexpectedly, relative transcripts of *LsGAS1/2* in stems were >200-fold higher than those in latex (Figure 2B). Among all the tissues examined, transcripts of *LsGAS1/2* were the lowest in the latex where their metabolic end-products accumulate. In contrast, *LsGAS3* showed ~2.5-fold higher expression in stems, relative to latex (Figure 2C). These results suggested that laticifers might not be the cellular sites for *LsGAS* expression.

### 2.2. Assessing LsGAS Transcription with Targeted RNA-Seq Analysis

The transcript levels of *LsGAS1-3* were independently analyzed by targeted Illumina read mapping analysis. Four individual lettuce plants were used to generate four replicates of Illumina-sequencing libraries either from the stem or latex. Using the libraries, 100 bp paired-end reads were carried out to obtain eight sequencing data sets, each of which had read numbers ranging between 55 to 69 million reads per library. Previously characterized biosynthetic genes for STLs (*LsGAS1/2/3, LsGAO1/2, LsCOS*), NR (*LsCPT3, LsCBP2*), and DHDD (*LaCPT1, LsCBP1*) [8,19,20,21] were used as reference templates to count read numbers mapped to these genes. Two isoforms of *FPS* (farnesyl diphosphate synthase), which provide a precursor for LsGAS, were also identified from the lettuce genome, and their expression in stem and latex was examined alongside *LsGAS1-3*. Consistent with the qPCR data, *LsGAS1* and *LsGAS2* showed two orders of magnitude higher levels (100–120-fold higher) of transcripts in stems than in latex (Table 1). The abundance of *LsGAS3* transcripts was significantly lower (40–70-fold lower) than those of *LsGAS1/2* with an almost negligible level of transcripts in stems. *LsGAS3* still showed higher expression in stems, but the difference between latex and stems was not statistically significant. Of the two *LsFPS* transcripts examined, *LsFPS2* showed negligible expression in latex and a 95-fold higher expression in stems, mirroring the expression pattern of *LsGAS1/2*. This suggests that LsFPS2 could be a major contributor for the substrate in STL biosynthesis, although more investigation is required. Similar to *LsGAS1/2*, the downstream biosynthetic genes, *LsGAO1* and *LsCOS*, also showed predominant expression in stems, closely resembling those of *LsGAS1/2*. However, the second isoform of *LsGAO* (*LsGAO2*) exhibited a 9.7-fold higher level of transcripts in latex than in stems.

In contrast to the expression of *LsGAS1/2*, the expression levels of NR-biosynthetic genes (*LsCPT3*/*LsCBP2*) were several folds higher in latex, relative to stems. These results are consistent with previously reported qPCR data and promoter-GUS analysis [8,13]. The transcripts for DHDD biosynthesis (*LsCPT1*/*LsCBP1*) were overall much less abundant than those for STL- and NR-biosynthesis and were present at comparable levels in latex and stems.

Based on these results, we concluded that *LsGAS1* and *LsGAS2*, the first enzymes committed to STL biosynthesis in lettuce, do not show any significant expression in latex where their metabolic end-products accumulate. These results are in sharp contrast to NR biosynthesis in lettuce. We previously showed that transcripts for NR-biosynthetic genes were mainly found in latex where NR accumulates [8,13].

### 2.3. Characterizing GAS1/2 Promoters with GUS-Fusion Constructs

Cell-type specific expression of *LsGAS1*/*2* was more precisely analyzed by characterizing their promoter activities. The *LsGAS1/2* promoters, each consisting of a ~1 kb fragment upstream of their ATG start codons, were identified from the lettuce genome. These promoters are hereafter referred to as *pLsGAS1* and *pLsGAS2*. The promoter fragments were transcriptionally fused to the reporter gene, β-glucuronidase (*GUS*). The promoter-GUS constructs and vector controls were individually transformed to lettuce, and a total of eight and twelve independent T1 lines were generated using the *pLsGAS1-GUS* and *pLsGAS2-GUS* constructs, respectively. Leaves from T1 lines were infiltrated with X-Gluc, and two lines for each construct that showed particularly strong GUS signals were selected to propagate the T2 generation for further analysis. 

In in situ GUS staining of T2 plants, the vector-transformed control showed no sign of GUS staining (Figure 3A), but both *pLsGAS1-GUS-* and *pLsGAS2-GUS*-transformed lettuces showed clear GUS activities along the vascular bundles (Figure 3B,C). However, as the GUS-staining patterns in the vascular bundles do not reveal the identity of specific cell-types in vascular bundles, the transgenic lettuces were fixed and microsectioned to precisely visualize the cells stained with GUS activity. In the microsections of vector control lettuce, laticifers were recognized by blue pigments (proteins stained by amido black 10B) and by characteristic cell fusions through degraded cell walls (yellow rectangles, Figure 3D). In contrast, microsections of *pLsGAS1-GUS* or *pLsGAS2-GUS* transgenic lettuces showed no evidence of GUS stains in the laticifers. Instead, negative laticifer images were observed over GUS-stained blue backgrounds in surrounding parenchyma cells (Figure 3E,F). The observed negative images of laticifers were in stark contrast to the intense GUS stains of laticifers in lettuce transformed with *LsCBP2* promoter *GUS-fusions* (*LsCBP2-GUS*) in our previous work [13]. To further confirm the lack of GUS activity in laticifers, latex samples were collected from vector control lettuce and from lettuces transformed with *pLsGAS1-GUS, pLsGAS2-GUS,* or *pLsCBP2-GUS* (a positive control for *GUS* expression in laticifers [13]). The in vitro latex assays showed no GUS activity in latex samples collected from the vector control and *pGAS1/2-GUS*-transformed lettuce (Figure 3G–I). However, intense GUS activity was detected in the latex collected from the *pLsCBP2-GUS*-transformed lettuce (Figure 3J). Taken together, we concluded that *LsGAS1/2* are not expressed in laticifers but are expressed in vascular parenchyma cells neighboring laticifers.

## 3. Discussion

This work’s key finding is that *LsGAS1/2* do not express in laticifers but in vascular parenchyma cells neighboring laticifers in lettuce. The plausible explanation for *LsGAS1/2* not being expressed in laticifers is to avoid precursor competition with NR-biosynthetic enzymes in laticifers. LsGAS1/2 and LsCPT3/LsCBP2 share a central precursor, IPP, for STL and NR biosynthesis, respectively. IPP is a building block for NR, and the molecular weight of NR in lettuce is known to reach 1.5–2.0 million g/mol [3,8,24], which implies that up to 30,000 IPP monomers are condensed to formulate a NR molecule in laticifers. Aside from NR, lettuce accumulates a copious amount of STLs in laticifers [2], and thus it would be a great challenge to consistently synthesize both NR and STLs in laticifers. One metabolic strategy to resolve this conflict is to express STL biosynthetic genes in the parenchyma cells connected to laticifers and transfer STL metabolites to laticifers after synthesis. 

Initially, qPCR data intrigued us to examine the cellular compartmentalization of STL- and NR-biosynthesis, but targeted RNA-Seq analysis provided more quantitative details of transcript levels of STL biosynthetic genes (Table 1). *LsGAS1/2* showed a markedly strong expression in stems with negligible levels of *LsGAS1/2* transcripts in laticifers. Considering that some non-laticifer cytosolic components could contaminate the latex samples when incisions were made in stems during sample collection, the transcript levels of *LsGAS1/2* could be even lower in laticifers. Similarly, two subsequent biosynthetic genes, *LsGAO1* and *LsCOS*, showed strong expression in stems. The simplistic view of early STL biosynthesis is that the sequential catalysis by LsGAS1/2, LsGAO1, and LsCOS funnels carbon flux from FPP to various STLs in the parenchyma cells. Although downstream enzymes remain uncharacterized, it is possible that the complete biosynthesis of STLs happens in the parenchyma cells adjacent to laticifers, followed by transport of the STLs to laticifers. 

The second isoform of *LsGAO* (*LsGAO2*) showed the opposite expression pattern, and significantly more *LsCOS* transcripts could be detected in laticifers in comparison to *LsGAS1/2* transcript levels in laticifers (Table 1). It appears that the expression of STL-biosynthetic genes becomes progressively stronger in laticifers after *LsGAS1/2*. However, to avoid precursor competition, only *LsGAS1/2* needs to be expressed in non-laticifer cells. Expression of the downstream biosynthetic genes, *LsGAO* and *LsCOS*, in non-laticifer cells is not necessary to avoid precursor competition, as their substrates are not IPP. Once the backbone of STLs, germacrene A, is synthesized from a unique IPP pool in the parenchyma cells, germacrene A and its downstream intermediates could be transported to laticifers for further transformation. 

In situ RNA hybridization and immunolocalization were previously performed in *Euphorbia tirucalli* and opium poppy, respectively, which found transcripts or enzymes in the parenchyma cells adjacent to laticifers [25,26]. However, those plants are not amenable for genetic transformation, limiting the uses of simple yet accurate promoter-*GUS* analysis through a transgenic approach. Present studies used the *pGAS-GUS*-transformed lettuce to provide a technically independent view of the contribution of vascular parenchyma cells to specialized metabolism in plants. Although we have not investigated the localization of LsGAS1/2 enzymes in this work, the complete lack of GUS activity in latex from the *pLsGAS1/2-GUS* transgenic lettuce suggests that protein transportation between cells is limited. Therefore, our model predicts that active metabolite trafficking operates from parenchyma cells to laticifers. We are currently working on identifying transporters responsible for STL movement in lettuce.

## 4. Materials and Methods

### 4.1. Plant Growth and DNA Isolation

Lettuce (cv. Ninja) was grown in a phytochamber at 22 °C for 16 h of light and 20 °C for 8 h of dark. Genomic DNA was isolated from leaves using the DNeasy Plant Mini Kit (Qiagen) and was used to amplify *LsGAS1/2* promoters.

### 4.2. Quantitative Real-Time PCR

Total RNA was extracted from various tissues using Trizol. Lettuce latex was isolated by making a small incision in stems with a razor blade. First strand cDNA was synthesized from total RNA using oligo(dT)_12–18_ and Superscript III reversed transcriptase (Invitrogen). qPCR was performed using the Step One Real-Time PCR System (Applied Biosystems) with primer sets 1/2, 3/4, 5/6 or 7/8 for *LsGAS1*-*3* and *Actin*, respectively. All primer sequences are listed in Appendix A. The PCR mixture (total 10 μL) was prepared by mixing cDNA (1 ng), primers (5 μM), and 5 μL of Power SYBR Green PCR Master Mix (Applied Biosystems). The qPCR program was set for 1 cycle of 95 °C for 10 min and 40 cycles of 95 °C for 15 s and 58 °C for 1 min. The relative transcript abundances were determined from the ΔCT values calculated using the reference gene (*actin*).

### 4.3. RNA-Seq Analysis

Lettuce latex and whole stem (including latex) samples were collected from four individual lettuce plants (2 months old). Total RNA was extracted using Trizol and additionally purified with the Illustra RNAspin Mini RNA Isolation kit (Fisher Scientific, Carlsbad, USA). Quality control of total RNA, library generation, and NovaSeq 6000 system (100 bp paired-end reads) Illumina sequencing were performed at Genome Quebec, McGill University, Canada. A total of eight sequencing data sets with a sequencing depth between 57.9 and 68.3 million reads per library were generated. The Map Reads to Contig function in CLC Genomics Workbench (version 21.0.4, Qiagen, Aarhus, Denmark) with a similarity fraction of 0.95 was used to count reads mapped to the reference genes listed in Table 1.

### 4.4. Plasmid Construction

The promoter sequences of *LsGAS1/2* were obtained from the Phytozome lettuce genome database. DNA fragments about 1 kb upstream of the *LsGAS1/2* start codon were amplified using primer sets of 9/10 or 11/12 (hybrid primers designed to include gene-specific sequences and common sequences for subsequent Gateway cloning), respectively, were cloned into the pGEMT-Easy vector (Promega, Madison, USA), and were sequenced. The *GAS* promoters were re-amplified with primer sets 13/14 (common primers for Gateway cloning) and cloned into pDONR221 (Invitrogen, Carlsbad, USA) using the Gateway BP Clonase II Enzyme mixture (Invitrogen). They were then cloned into the destination vector pKGWFS7 [27] with the Gateway LR Clonase II Enzyme mixture (Invitrogen).

### 4.5. Generating Transgenic Plants, Histochemical GUS Staining, and Microsections

Transgenic lettuce plants were generated as described [13]. GUS activity in T2 transgenic lettuce was detected according to the methods previously described [13,28]. Leaves of 1–2 week old transgenic plants were submerged and infiltrated with a GUS staining solution (10 mM EDTA, 0.1% Triton–X 100, 1 mM K_3_Fe(CN)_6,_ 2 mM X-Gluc substrate, and 0.1 M Na_3_PO_4,_ pH 7) under vacuum for 2–3 h. The leaves were then incubated at 37 °C for 16–18 h. Leaves were cleared using an ethanol series beginning with 50% ethanol and increasing in increments of 5% every 12 h. Leaf staining patterns were observed using an Olympus SZX2-ILLT light microscope. Plants with strong expression were grown to an early bolting stage (3 weeks), and 1 mm thick longitudinal hand sections were prepared from their stems. GUS staining in stems was performed as described for leaves. After staining, the stems were fixed using a fixative solution (1.6% paraformaldehyde, 2.5% glutaraldehyde, and 0.05 M phosphate buffer, pH 6.8). The samples were fixed for 1–2 weeks. After fixing, the samples were dehydrated in 100% ethanol and embedded with Technovit 7100 (Electron Microscopy Sciences, Hatfield, USA). The samples were sectioned using Ralph knives and a Leica Reichert-Jung 2040 Autocut rotary microtome to a thickness of 3 µm. Slides were examined using a Leitz Aristoplan microscope (Leitz). Images were generated using a Nikon Digital Sight, DS-Fi2 camera (Nikon).

### 4.6. Metabolite Analyses

STLs and NR from lettuce latex were measured, according to the published methods [8,21].

## Figures and Tables

**Figure 1 plants-11-01192-f001:**
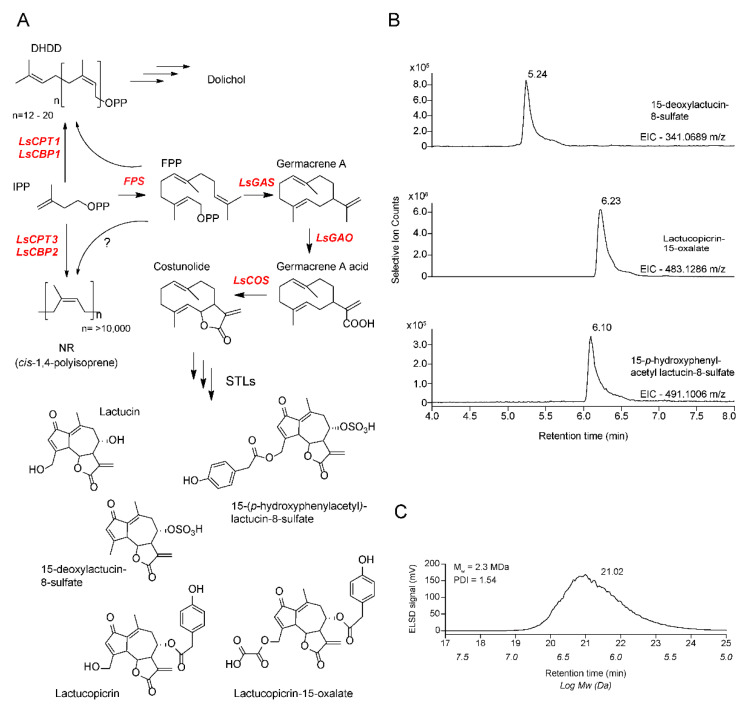
Biosynthetic pathways and metabolite analysis of NR and STLs in lettuce latex. (**A**) Biosynthetic routes of diverse secondary isoprenoids in lettuce latex. (**B**) STL detection in lettuce latex by LC-qTOF. [M+H]^+^ ions were detected in positive ion mode. (**C**) NR detection in lettuce latex by HPLC-GPC (gel permeation column)-ELSD (evaporative light scattering detector). Abbreviations used are: DHDD, dehydrodolichyl diphosphate; STL, sesquiterpene lactone; NR, natural rubber; IPP, isopentenyl diphosphate; FPP, farnesyl diphosphate; PDI, polydispersity index. The key enzymes for each biosynthetic pathway are labeled in red. Representative STL products are shown. Schemes follow the same formatting.

**Figure 2 plants-11-01192-f002:**
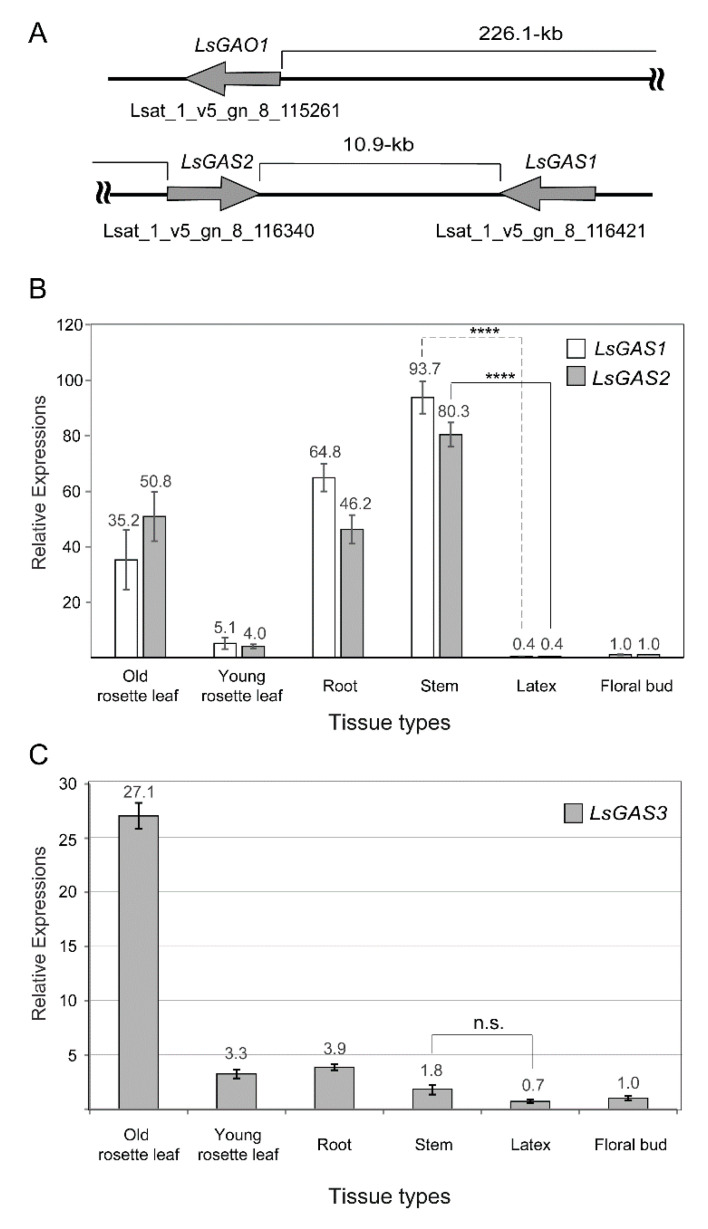
Genomic cluster and expression analysis of *LsGAS*. (**A**). Schematic representation of *LsGAS1*, *LsGAS2*, and *LsGAO1,* retrieved from Phytozome. (**B**,**C**). qPCR analyses of *LsGAS1*/2 (**B**) and *LsGAS3* (**C**) in various lettuce tissues. Young rosette leaves were collected 10–14 days after germination; old rosette leaves were collected 4–6 weeks after germination. The transcript level in floral buds was set to one, and all other values were relative to the floral bud transcript level. Stem samples used whole stem tissues including latex. Data are means ± S.D. (*n* = 4). **** indicates a *p* value of < 0.0001; n.s. indicates no statistical significance, *p* value >0.05.

**Figure 3 plants-11-01192-f003:**
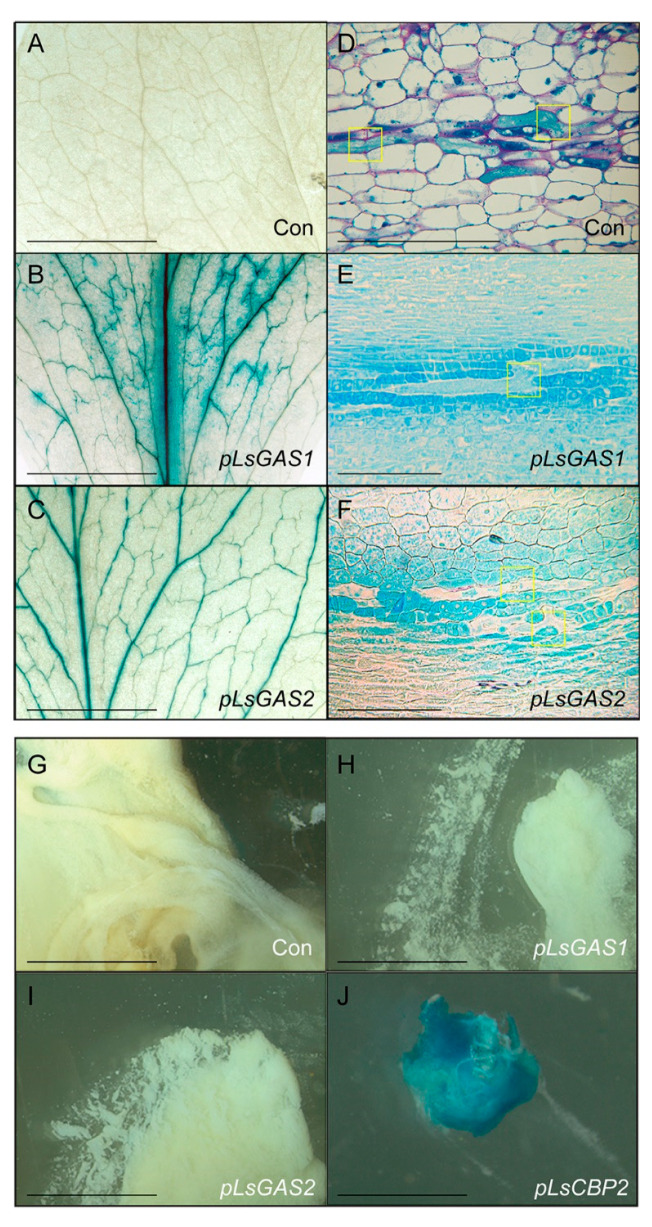
*LsGAS1* and *LsGAS2* promoter analyses in transgenic lettuce. (**A**–**C**). In situ GUS-staining patterns in leaves of transgenic lettuce. Scale bars are 5 mm. (**A**) Vector control (Con); (**B**) *pLsGAS1-GUS*-transformed lettuce; (**C**) *pLsGAS2-GUS*-transformed lettuce. (**D**–**F**) Microsections of stems at early bolting stage from wild-type control (Con) and transgenic lettuce (*pLsGAS1/2-GUS* constructs). Scale bars are 20 µm. (**D**) The high density of proteins in laticifers were stained with amido black 10B, and total carbohydrates were stained with periodic acid-Schiff’s (PAS) stain in vector control lettuce. Yellow rectangles show characteristic features of cell fusions in laticifers. (**E**,**F**) GUS activities are visualized by blue pigment. Laticifers with cell-fusions (indicated by yellow rectangles) are shown as negative images in blue backgrounds. (**G–J**) GUS activity assays using isolated latex. Previously reported transgenic lettuce (*pLsCBP2-GUS*) was used as a positive control [13], and vector control was used as a negative control (Con). Scale bars are 5 mm.

**Table 1 plants-11-01192-t001:** Quantitative read maps for transcripts in STL-, NR-, and DHDD-biosynthesis in lettuce latex and whole stems, including latex.

Target Genes	^1^ Metabolic Products	Stem— ^2^ Mapped Read Number	Latex— Mapped Read Number	^4^ Fold (Stem/ Latex)
*LsGAS1*	STL	^5^ 311.5 ± 95.5	2.6 ± 1.1	** 119.81
*LsGAS2*	STL	521.7 ± 176.6	5.0 ± 1.3	* 104.34
*LsGAS3*	STL	7.5 ± 8.9	0.1 ± 0.2	75.00
*LsGAO1*	STL	1375.1 ± 343.3	8.5 ± 2.4	** 161.78
*LsGAO2*	STL	9.5 ± 4.1	92.5 ± 23.4	** 0.10
*LsCOS*	STL	746.9 ± 124.4	72.2 ± 29.4	* 10.34
*LsCPT3*	NR	312.4 ± 38.7	2251.7 ± 259.3	** 0.14
*LsCBP2*	NR	337.8 ± 51.5	2664.2 ± 423.0	** 0.13
*LsCPT1*	DHDD	17.7 ± 1.1	12.6 ± 1.1	** 1.40
*LsCBP1*	DHDD	9.3 ± 0.8	9.7 ± 0.6	0.96
^3^ *LsFPS1*	FPP	528.8 ± 122.0	1537.1 ± 164.4	** 0.34
^3^ *LsFPS2*	FPP	95.7 ± 25.7	1.0 ± 0.4	** 95.70

^1^ The abbreviations are as follows: STL, sesquiterpene lactone; NR, natural rubber, DHDD, dehydrodolichyl diphosphate; FPP, farnesyl diphosphate. ^2^ The number of the reads mapped to a targeted transcript per million reads. ^3^ *LsFPS1* and *LsFPS2* are *Lsat_1_v5_gn_5_2060.1* and *Lsat_1_v5_gn_7_114941.1*, respectively, in the lettuce genome. ^4^ * indicates a *p* value of < 0.05 and ** indicates a *p* value of < 0.01. ^5^ Data are average ± SD (*n* = 4). Each data set is from independently prepared Illumina libraries from different lettuce plants.

## Data Availability

The genome sequences of lettuce *GAS* isoforms used in our study were downloaded from the Phytozome (Available online: http//www.phytozome-next.jgi.doe.gov, accessed on 14 April 2022).

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
