# Peer review of "Germacrene A Synthases for Sesquiterpene Lactone Biosynthesis Are Expressed in Vascular Parenchyma Cells Neighboring Laticifers in Lettuce"

_plants, 2022, doi:10.3390/plants11091192_

Round 1
Reviewer 1 Report
The text bellow contains comments on manuscript entitled “Germacrene A synthases for sesquiterpene lactone biosynthesis are expressed in vascular parenchyma cells neighboring laticifers in lettuce”.
The manuscript is focused on expression studies of germacrene A synthase (LsGAS), which catalyzes the first step in STL biosynthesis in lettuce.
To my opinion, the manuscript is well written with a logically structured experimental design. I think that the authors have sufficient proves for the change in LsGAS expression by qPCR and GUS expression.
In case the authors consider that it might be of interest and would improve the quality of the work they might include chemical analysis of sesquiterpene lactones and/or natural rubber, which biosynthesis is expected to be affected.
Author Response
Question - The manuscript is focused on expression studies of germacrene A synthase (LsGAS), which catalyzes the first step in STL biosynthesis in lettuce. To my opinion, the manuscript is well written with a logically structured experimental design. I think that the authors have sufficient proves for the change in LsGAS expression by qPCR and GUS expression. In case the authors consider that it might be of interest and would improve the quality of the work they might include chemical analysis of sesquiterpene lactones and/or natural rubber, which biosynthesis is expected to be affected.
Answer - We thank this reviewer’s constructive comments. Although metabolites (natural rubber and sesquiterpene lactones) were previously measured (reference 2 and 3 in the manuscript), according to this reviewer’s suggestion, we re-measured the natural rubber and sesquiterpene lactones by LC-qTOF and HPLC-ELSD. The new data is now included in Figure 1B/C and new sentences describing this data in the Result. We believe the readability of this manuscript increased after adding this new data.
Reviewer 2 Report
The manuscript entitled “Germacrene A synthases for sesquiterpene lactone biosynthesis are expressed in vascular parenchyma cells neighboring laticifers in lettuce” describes the expression pattern of STL genes and provides an interesting study regarding possible metabolite or intermediate transport between different cell types. However, the amount of data is and their presentation quality is rather low. I would recommend to include some more data (see details in comments below) and an improvement of data presentation is necessary before I would recommend the manuscript for publication.
Major points:
- If the authors want to present statements/conclusions regarding intermediate metabolite-trafficking of the STL biosynthesis, they should include qPCR analysis and/or promoter-GUS studies not only for the GAS-genes, but also for the genes which are involved in the intermediate pathway steps (GAO and COS). Alternatively, if intermediate metabolites are measurable/ detectable, the authors should quantify the intermediates in the different tissues.
- Figure 2 A: If the authors want to draw conclusions regarding STL pathway gene location, they should present the gene location in the different linkage group also for GAS3, as well as for GAO and COS.
- Figure 2 C/D: I would recommend to present the qPCR data using absolute values (not relative ones) and to show all three genes (in case of GAS) in one figure. Furthermore, the authors should include statistical analysis in the expression analysis.
- Figure 3: Please include scale bars in the figure. A, D and G do not show a WT-sample but an empty vector control. Please change the labelling accordingly. From which tissue/part of plant are the microsections in D-F? Please specify in the figure legend.
- Does the stem tissue, which was used for qPCR and RNAseq-analysis, contain latex or do the authors harvest “latex-free stem tissue”? Please specify in the figures and in material and method section.
Minor points:
- Line 47: change “to” to “and”
- Line 64: the isoforms LsCPTL2 and LsCPT3 do not form the protein complex in dandelion. Please add an extra sentence and describe in more detail the protein isoform in dandelion.
- Line 261: for which analysis genomic DNA was isolated?
Author Response
We appreciate the detailed review from this reviewer. In the letter below, we made our best efforts to address the raised critiques in given time.
Major points:
Comment #1 - If the authors want to present statements/conclusions regarding intermediate metabolite-trafficking of the STL biosynthesis, they should include qPCR analysis and/or promoter-GUS studies not only for the GAS-genes, but also for the genes which are involved in the intermediate pathway steps (GAO and COS). Alternatively, if intermediate metabolites are measurable/ detectable, the authors should quantify the intermediates in the different tissues.
We agree that additional “promoter-GUS” and “qPCR analysis” of other genes can offer a more comprehensive view of STL biosynthesis. However, the suggested promoter-GUS experiments will take 12- 18 months as new sets of transformation are required. Thus, making additional transgenic lettuce is beyond the scope of this project.
For qPCR data, we conducted the qPCR of LsGAS1-3 for an initial discovery, which help us formulate the hypothesis of this manuscript. After the initial discovery, we invested our time and resources to the Illumina sequence-based quantitative transcriptome described in Table 1. Sequencing-based transcript analysis is far simpler, leaving much less room for technical mistakes. Thus, the data in Table 1 give us quite accurate data for GAS/GAO/COS transcripts, which addresses the requested qPCR experiments in my ipinion.
While we are measuring STLs to address the reviewer 1’s request (see revised Fig 1 B/C) by LC-MS, we also attempted to detect intermediates in latex. However, we could not detect any intermediates (i.e., germacrene A, germacrene A acid, and costunolide) in our LC-MS analysis in latex, whereas final STL products were clearly detected in latex (Fig 1B/C). Intermediates may not accumulate due to their transient nature in the metabolic pathways (intermediates are likely to be rapidly converted to next compound in the pathway).
Latex is relative pure samples from laticifers (cytoplasmic content of laticifers), and laticifers are tightly imbedded in vasculatures throughout all tissues in lettuce (stems, roots, leaves, and even flower petals). It is not possible to isolate tissues without laticifers in lettuce. Therefore, if we measure STLs from different tissues, we are not able to tell whether STLs are synthesized in different cells (non-laticifer cells) or laticifers present in different tissues. Due to this inherent problem, we wish to only focus on metabolite analysis in latex.
Comment #2 - Figure 2 A: If the authors want to draw conclusions regarding STL pathway gene location, they should present the gene location in the different linkage group also for GAS3, as well as for GAO and COS.
Thank you for this comment. We analyzed the genomic locations of GAS3, GAO1/2 and COS in the published lettuce genome. We found that LsGAO1 is linked with LsGAS1/2 in the same chromosome. However, other genes are not linked, and they are scattered throughout the genome. Revised Figure 2A is now included in the new manuscript.
Comment #3 - Figure 2 C/D: I would recommend to present the qPCR data using absolute values (not relative ones) and to show all three genes (in case of GAS) in one figure. Furthermore, the authors should include statistical analysis in the expression analysis.
Statistical analysis with an emphasis on expression levels in stem and latex is included in the revised Figure 2B/2C.
Thank you for the suggestion for absolute qPCR. We feel that quantitative transcript data given in Table 1 provide sufficient transcript data (direct quantitative comparison of different genes), which properly addressed this comment. The qPCR analysis aiming to obtain absolute quantitation is prone to technical mistakes, while there is less room for mistakes in sequence-based transcript analysis. Thus, we believe the data in Table 1 addressed this comment. Also, please see my explanation for comment #1.
Comment #4 - Figure 3: Please include scale bars in the figure. A, D and G do not show a WT-sample but an empty vector control. Please change the labelling accordingly. From which tissue/part of plant are the microsections in D-F? Please specify in the figure legend.
Scale bars are now added in Figure 3. Controls were labeled with "Con" in the figures and additional descriptions are added in the legends for the control and the tissues used for microsections.
Comment - Does the stem tissue, which was used for qPCR and RNAseq-analysis, contain latex or do the authors harvest “latex-free stem tissue”? Please specify in the figures and in material and method section.
We used “whole stems including latex” as it is practically impossible to prepare “latex-free-stem tissue”. According to the suggestion, in the Figure/Table legend and method & Materials, we specify this information.
Minor points:
- Line 47: change “to” to “and”
Corrected as suggested.
- Line 64: the isoforms LsCPTL2 and LsCPT3 do not form the protein complex in dandelion. Please add an extra sentence and describe in more detail the protein isoform in dandelion.
We revised the sentence to clarify the information.
".... a protein complex is formed by cis-prenyltransferase isoform 3 (LsCPT3) and CPT-Like protein isoform 2 (LsCPTL2) on the cytosolic side of the endoplasmic reticulum (ER) in lettuce (Figure 1A), and its related species dandelion also has orthologous proteins that form a protein complex on ER [8,11]."
Line 261: for which analysis genomic DNA was isolated?
Additional information was included to clarify this comment - "....... and used to amplify LsGAS1/2 promoters."
Author Response
Comment 1 - What was the pattern of gene(s) encoding farnesyl-PP synthase(s) with different cell-type specific presence? Should this data be available, the inclusion into the manuscript would underline the general observations.
Answer - Thank you for this constructive suggestion. As suggested, we conducted quantitative transcript mapping for farnesyl-PP synthase (FPS). First, lettuce genome search identified two isoforms of FPS (LsFPS1 and LsFPS2). Second, quantitative transcript mapping showed these two FPSs showed different expression patterns in latex and stem. Of particular interest is that LsFPS2 transcripts are 95-fold more abundant in stem in comparison to those in latex. This suggest that one of the isoforms (LsFPS2) could be a major (or exclusive) player for sesquiterpene lactone biosynthesis, although further experiments are required to prove this hypothesis. We included this new data in Table 1 and also includes sentences describing this result in the Result. We believe this new data improve the quality of this manuscript and thank you for the suggestion.
Comment 2 - If you follow the rule of writing names of genes in a 3-letter code, then it is in italics, but here you speak of enzymes, thus NOT written in italics!
Answer - Corrected as suggested. Italic fonts were converted to regular fonts to describe proteins (enzymes).